# The New Buffer Salt-Protected Sodium Butyrate Promotes Growth Performance by Improving Intestinal Histomorphology, Barrier Function, Antioxidative Capacity, and Microbiota Community of Broilers

**DOI:** 10.3390/biology13050317

**Published:** 2024-05-01

**Authors:** Mebratu Melaku, Dan Su, Huaibao Zhao, Ruqing Zhong, Teng Ma, Bao Yi, Liang Chen, Hongfu Zhang

**Affiliations:** 1State Key Laboratory of Animal Nutrition and Feeding, Institute of Animal Science, Chinese Academy of Agricultural Sciences, Beijing 100193, China; sefibahir2009@gmail.com (M.M.); sudanfir@163.com (D.S.); huaibao82@163.com (H.Z.); zhongruqing@caas.cn (R.Z.); chenliang01@caas.cn (L.C.); zhanghongfu@caas.cn (H.Z.); 2Department of Animal Science, College of Agriculture, Woldia University, Woldia P.O. Box 400, Ethiopia

**Keywords:** sodium butyrate, growth performance, intestinal health, short-chain fatty acids, cecal microbiota, broilers

## Abstract

**Simple Summary:**

The misuse and overuse of antibiotics in food animal production has brought about an antibiotic resistance crisis in the 21st century. To fight against this silent global pandemic through the transmission of poultry production, various in-feed antibiotic alternatives, primarily sodium butyrate products, are under evaluation, and promising results have been obtained in previous studies. To enhance this green feed additive evolution, this study evaluated the effects of a new type of buffer salt-protected sodium butyrate (NSB), which uses buffer salts to protect sodium butyrate, on the growth performance, various intestinal health indicators, and cecum microbiota of broilers during the rapid growth stage. The result shows that NSB improves growth performance, serum anti-inflammatory cytokines, gut morphology, intestinal immunity and antioxidant capacity, short-chain fatty acids’ (SCFAs’) content, and cecum microbiota, indicating that NSB can be a potential additive supporting the green feed additive industry in broiler nutrition.

**Abstract:**

In this study, a commercial sodium butyrate protected by a new buffer salt solution (NSB) was tested to determine whether it can be used as an antibiotic alternative in broiler production. A total of 192 1-day-old broilers were randomly allocated to three dietary treatments: soybean meal diet (CON), antibiotic diet (ANT, basal diet + 100 mg/kg aureomycin), and NSB (basal diet + 800 mg/kg NSB). The growth performance, serum anti-inflammatory cytokines, intestinal morphology, gut barrier function, antioxidative parameters, SCFAs’ content, and cecal microbiota were analyzed. The result showed that NSB significantly improved ADFI and ADG (*p* < 0.01), and decreased FCR (*p* < 0.01). Serum anti-inflammatory cytokine IL-10 was up-regulated (*p* < 0.01), and pro-inflammatory TNF-α was down-regulated (*p* < 0.05) by NSB supplementation. H&E results showed that VH and the VH/CD ratio significantly increased (*p* < 0.05) in the jejunum and ileum in the NSB group. Furthermore, ZO-1 (*p* < 0.01), claudin-1 (*p* < 0.01), and occludin (*p* < 0.05) in the jejunum and claudin-1 (*p* < 0.01) and mucin-2 (*p* < 0.05) in the ileum were significantly up-regulated in the NSB group. Additionally, SOD (*p* < 0.05) and the T-AOC/MDA ratio (*p* < 0.01) in the jejunum and SOD in the ileum were significantly increased (*p* < 0.05) in the NSB group. The MDA level also significantly increased (*p* < 0.01) in the ANT group in the jejunum. Propionic acid (*p* < 0.05) and butyric acid (*p* < 0.01) content significantly increased in the NSB group in the jejunum and ileum segments. The 16S rRNA sequencing results showed no significant difference (*p* > 0.05) in alpha and beta diversity among the groups. LEFSe analysis also indicated that *Peptostreptococcaceae*, *Colidextribacter*, *Firmicutes*, *Oscillospira*, and *Erysipelatoclostridiaceae*, which promote SCFA production (*p* < 0.05), were identified as dominant taxon-enriched bacterial genera in the NSB group. The Spearman correlation analysis revealed that *Colidextribacter* with ADFI, ADG, VH, claudin-1 (*p* < 0.05), and *unclassified_f__Peptostreptococcaceae* with ADFI, IL-10, and ZO-1 were positively correlated (*p* < 0.05). Furthermore, ADFI and ADG with IL-10, claudin-1, SOD, T-AOC, and butyric acid (*p* < 0.05), and similarly, ADG with VH (*p* < 0.05), showed a positive correlation. In conclusion, NSB enhanced the growth performance by improving jejunum and ileum morphology, and serum anti-inflammatory cytokines, and by regulating the intestinal barrier function and antioxidant capacity, SCFAs’ content, and cecum microbiota, showing its potential use as an alternative to antibiotics in poultry nutrition.

## 1. Introduction

Poultry production is an emerging livestock industry that plays a significant role in meeting the increasing demand for animal protein consumption of the rapidly growing human population [1]. However, the intensive poultry industry has encountered a series of problems, for example, housing environmental stress and affected healthy breeding, especially when antibiotics are banned in animal feeds as growth promoters in many countries.

Various feed additives have been extensively studied as alternatives to antibiotics in food animal production. Sodium butyrate (SB), in both coated and uncoated forms, has emerged as a potential feed additive due to its ability to quickly convert into butyric acid in the birds’ intestines and improve gut health through several mechanisms. It reduces harmful bacteria colonization by lowering intestinal pH, influences the morphology of the digestive and lymph organs, and enhances feed efficiency and weight gain [2]. SB primarily acts as an energy source for epithelial cells, contributing to the development of gut wall tissue, maintenance of intestinal integrity, enhancement of immunity, modulation of intestinal microbiota, and increase in short-chain fatty acids’ (SCFAs’) concentration [3,4]. Multiple studies have investigated the effects of different types of SB products as potential growth promoters and disease treatments in chickens and reported variable results [5,6].

Mátis et al. [7] demonstrated that dietary-protected SB positively influenced SCFAs’ content, gut health, and gut barrier integrity in broilers at 21 days of age. Previous studies conducted in our laboratory also showed that chemically protected sodium butyrate (CSB) increased the concentrations of butyric acid and total SCFAs in the jejunum and cecum, improved the structural integrity of the intestine, and positively affected the dynamics of cecal microflora in broilers [8,9,10]. However, Wu et al. [11] found no significant improvements in broiler growth performance, while Wan et al. [9] highlighted the beneficial effects of protected SB compared to the long-term impacts of lincomycin antibiotics. Recently, a new product called buffer salt-protected SB (NSB) has been developed to prevent early dissociation at low pH in the stomach and ensure sufficient release of butyrate in the chicken intestine (Fujian Alliance Biotechnology Co., Ltd., Sanming, China). This new product has not yet been evaluated under experimental conditions in broilers.

As a result, previous studies have also not investigated the serum inflammatory cytokine, morphological structure intestinal gut barrier function and oxidative stress, SCFAs’ content, and cecal microbiota profile, which are essential indicators of broiler performance and gut health during the rapid growth stage. Additionally, chickens do not have fully developed digestive systems at an early age, so buffer salt-protected SB may not be fully digested and absorbed in the intestine. Furthermore, the composition of gut microbiota in broilers undergoes significant changes during the rapid weight gain phase, while it remains relatively stability during the rapid skeletal growth phase, highlighting the distinct variations across different stages of development [12]. Therefore, this research aims to investigate how the NSB affects growth performance, serum cytokines and immunoglobulins, histomorphology, intestinal immunity and oxidative capacity, SCFAs’ content, and cecal microbiota during the rapid developmental stage of broilers.

## 2. Materials and Methods

### 2.1. Diet, Ethical Approval, Design of Experiment, and Chicken Management

NSB, a new product protected by buffer salts, was provided by Fujian Alliance Biotechnology Co., Ltd., Sanming, China. The chicken experiment was approved by the Animal Ethics Committee of Experimental Animal Welfare and Ethical Procedures of the Institute of Animal Science, Chinese Academy of Agriculture Sciences (IAS2021-105).

A total of 192 1-day-old Arbor Acres (AA) male broiler chickens having an initial body weight of 45.3 ± 0.18 g were randomly distributed into three treatment groups, each with eight replications and eight chicks per replicate, with a completely randomized design (CRD). Diets included a corn–soybean meal diet used as a control (CON), a diet added with antibiotics (CON + 100 mg/kg aureomycin, ANT), and a CON diet added with new SB (NSB) (CON + 800 mg/kg NSB). The basal diets of broilers were formulated to meet the protein (iso-nitrogenous, CP in %) and metabolizable energy (iso-caloric, ME kcal/kg DM) requirements recommended by the National Research Council (NRC, 1994) and China feeding standard of chicken (China NY/T 33–2004, 2004) (Table 1). All of the chicken management practices followed the management regulations of the State Key Laboratory of Animal Nutrition and Feeding, Institute of Animal Science (IAS), Chinese Academy of Agricultural Sciences (CAAS). Dosage references for the antibiotics (aureomycin, 100 mg/kg) and NSB (800 mg/kg) were taken from other SB product studies [8,11].

### 2.2. Slaughter and Sample Collection

At 42 days old, eight broilers per treatment (one broiler per replication) were starved for a 12 h fasting period and slaughtered with the exsanguination method by cutting the jugular vein. Blood samples were taken, and the serum was centrifuged at 3000 rpm per minute for 10 min and put at −20 °C to identify inflammatory cytokines and immunoglobulin indicators. Jejunum and ileum segments were fixed in a 4% paraformaldehyde solution for histomorphological examination. Similarly, jejunum and ileum tissue and digesta samples were collected and kept at −80 °C to measure intestinal immunity and antioxidant capacity and SCFAs’ content. Similarly, the cecum content was also used for 16S rRNA sequencing analysis to determine the cecal microbiota dynamics of broilers.

### 2.3. Growth Performance

Weekly body weight and feed intake measurements were taken, and the average daily weight gain (ADG), average daily feed intake (ADFI), and feed conversion ratio (FCR) were determined.

### 2.4. Serum Cytokine Expression and Immunoglobulins Indicators

The serum expression of pro-inflammatory cytokines such as tumor necrosis factor alpha (TNF-α) (cat# 88-7346), interleukin-1β (IL-β) (cat# 88-7261), and interleukin-6 (IL-6) (cat# 88-7066), and anti-inflammatory cytokines such as interleukin 10 (IL-10) (cat# 88-7106), as well as blood immunoglobulin indicators such as IgA, IgG, and IgM (HY-753), were determined using ELISA kits from Thermo Fisher Scientific Inc. Vienna, Austria according to the manufacturer’s instructions.

### 2.5. Analysis of Small Intestinal Histomorphology

The jejunum and ileum samples were fixed with paraformaldehyde (4%) and dehydrated with alcohol (70, 80, and 90%). The samples were then cleaned using xylene solution and embedded in paraffin wax. After that, samples were sliced at a 5 µm thickness using a microtome (KD-3358 Semi-automated Rotary microtome, Zhejiang Jinhua Kedi Instrumental Equipment Co., Ltd., Jinhua, China). The samples were stained with the hematoxylin and eosin (H&E) method, and the histological sections were examined under a light microscope (Leica Microsystems (SEA), Pte Ltd., SG-608924 Singapore) enlarged 5–10× using Leica Application Suite (LAS X) software, version 4.9 (Leica Microsystems CMS, GmbH, Heerbrugg, Switzerland). The villus height (VH) and crypt depth (CD) were measured in eight replicates for each VH and CD section, and finally, the VH to CD ratio was calculated.

### 2.6. Intestinal Gut Barrier and RT-qPCR

Total RNA was extracted from the jejunum and ileum mucosal tissue with TRIzol^®^ reagent (Life Technologies, 15596018, Carlsbad, CA, USA) following the manufacturer’s protocol. The quality (A260/A280 absorbance ratio between 1.8 and 2.00) and concentration of the pure RNA were checked twice using a NanoDrop 2000c spectrophotometer (Nanodrop technologies, Wilmington, NC, USA). Then, leveled RNA samples were transcribed to cDNA using Prime Script™ RT reagent kits with gDNA eraser (TaKaRa, RR047A, Dalian, China). The cDNA samples were diluted, and RT-qPCR was performed using TaKaRa TB Green^®^ Premix Ex Taq ^TM^ (Tli RNaseH plus, Takara Biomedical Technology Co., Ltd, Beijing, China) reagent kits following the instructions on the ABI Q7 Flex real-time PCR system (ABI, Singapore). The RT-qPCR was amplified in three stages: pre-denaturation, one cycle at 95 °C for 30 s; denaturation, 40 cycles at 95 °C for 5 s; and melting/dissociation curve, one cycle at 60 °C for 30 s. Four common pairs of primers were designed to detect gut barrier and tight junction protein genes (Appendix A), which are more indicators of intestinal inflammation and gut barrier function/integrity. β-actin was applied as an internal reference or a housekeeping gene, and the relative expression of each gene was calculated by using the 2^^−ΔΔCT^ method [13].

### 2.7. Spectrophotometric Determination of Antioxidant Enzyme Activity

Jejunum and ileum mucosa tissue samples (0.1 g) with 0.9 mL of NaCl solution (1:9 ratio) were homogenized for 1–2 min using a tissue homogenizer (SCIENTZ-48, Ningbo Xinzhi Biotechnology Co., Ltd., Ningbo, China). After 5 min at room temperature, the homogenized samples were centrifuged at 3000 rmp for 10 min at 4 °C, and 0.5 mL of the surface supernatant mucosa samples was taken for total protein and antioxidant enzyme determination. The total protein concentration of each sample was determined following the instructions using the Pierce^TM^ BCA (Bicinchoninic Acid) protein assay kit (Ref. 23227, Protein Biology, Thermos Scientific, Rockford, IL, USA). Indexes of superoxide dismutase (SOD) (cat#A001-3), total antioxidant capacity (T-AOC) (cat# A015-1), and malondialdehyde (MDA) (cat#A003-1) were determined based on the instructions of the commercial kits (Nanjing Jiancheng Bioengineering Institute, Nanjing, China). Moreover, the ratio of T-AOC to MDA (T-AOC/MDA) was calculated to evaluate the antioxidant/oxidative balance of the mucosa tissues.

### 2.8. Short-Chain Fatty Acids’ (SCFAs’) Analysis

The gas chromatography method was used to determine the contents of short-chain fatty acids including acetic, propionic, and butyric acids. In general, a 0.5 g sample from the jejunum and ileum contents was taken, and a mixture of an aliquot of the supernatant fluid and 25% metaphosphoric acid solution was used to create the standard curve. After remaining in a 1.5 mL centrifuge tube at 4 °C for more than 2 h, 0.1 mL was vortexed for 1 min and centrifuged at 1000 rpm for 10 min at 4 °C with a 0.9:0.1 mL ratio. Then, the supernatant was filtered through a 0.45 µm Milled-LG filter (Millipore, Billerica, MA, USA) and analyzed using the Agilent 7890N gas chromatograph (Agilent, Santa Clara, CA, USA).

### 2.9. Cecal Microbiota Analysis Using 16S rRNA Amplicon Sequencing

A total of 24 cecal samples were used for the total microbial genomic DNA extraction with three repeats using the AxyPrepDNA gel recovery kit (AXYGEN Biosciences, Union City, CA, USA). The same samples used for PCR were combined and identified using 2% agarose gel electrophoresis. PCR products were detected and quantified using the QuantiFluor dsDNA System kit (Promega Corporation, Madison, WI, USA). The hypervariable region V3-V4 of the bacterial 16S rRNA gene was amplified with primer pairs 338F_806R (5′ACTCCTACGGGAGGCAGCAG-3′ and 3′GGACTACHVGGGTWTCTAAT-5′) using an ABI GeneAmp^®^ 9700 PCR thermocycler (MARSHAL Scientific, Hampton, NH, USA).

According to the standard protocols of Majorbio Bio-Pharm Technology Co., Ltd. (Shanghai, China), purified amplicons were pooled in equimolar amounts to construct Illumina paired-end libraries. The quantity and size of the amplicon library were determined using an Illumina Library Quantification Kit (Kapa Biosciences, Woburn, MA, USA) and an Agilent 2100 Bioanalyzer (Agilent, Santa Clara, CA, USA). The libraries were sequenced on an Illumina MiSeq PE300 platform/NovaSeq PE250 platform (Illumina, San Diego, CA, USA), as mentioned by Zhou et al. [14]. The raw sequencing reads were deposited in the NCBI Sequence Read Archive (SRA) database (Accession Number: PRJNA1025322).

### 2.10. Statistical and Bioinformatics Data Analysis

Data on growth performance, serum cytokines and immunoglobulins, intestinal morphology, intestinal immunity, antioxidant capacity, and SCFAs’ content were analyzed using one-way ANOVA with the Tukey mean comparison test and presented with mean ± SEM. SPSS (version 27) and GraphPad (version 9.3.1) software were used for the data analysis and graphics. Raw sequencing data and bioinformatic analysis of the cecal microbiota, as well as the Spearman rank correlation coefficient, were performed using different software packages on the Majorbio Cloud platform (https://www.majorbio.com/; accessed on 1 June 2023) and OriginPro^®^ (version 2022) software. Operational taxonomic units (OTUs) information was used to calculate alpha diversity indexes with Mothur software (version 1.30.2, https://www.mothur.org/wiki/Download_mothur; accessed on 1 June 2023), and OTU similarity level 97% (0.97) was selected for analysis with Uparse software (version 11, http://www.drive5.com/uparse/; accessed on 1 June 2023). The Venn diagram analysis and graphing were performed using R-language (version 3.3.1). Principal co-ordinate analysis (PCoA) was analyzed using the R-Vegan v2.5-3 package. Linear discriminant analysis (LDA) effect size (LEfSe) (LDA score > 2, *p* < 0.05) was analyzed to determine the significant abundant taxa (genera to OTU level) of bacteria among the three groups. Mean separation was conducted for treatments with a significant difference with *p* < 0.05.

## 3. Results

### 3.1. The New Buffer Salt-Protected Sodium Butyrate (NSB) Improved Growth Performance of Broilers

The growth performance of broilers is shown in Table 2. ADG and ADFI were significantly increased (*p* < 0.001) in the NSB compared to the CON and ANT groups. However, the CON and ANT groups showed no significant difference (*p* > 0.05). Similarly, FCR was significantly decreased (*p* < 0.01) in the NSB compared to the other two groups. Conversely, the CON and ANT groups had no significant difference (*p* > 0.5).

### 3.2. The NSB Altered Serum Inflammatory Cytokines

The serum expression of pro-inflammatory and anti-inflammatory cytokines and serum immunoglobulins is shown in Figure 1A,B. IL-10 cytokine significantly increased (*p* < 0.001) in the NSB group compared to the CON and ANT groups. Conversely, INF-α was significantly up-regulated (*p* < 0.05) in the ANT group compared to the CON and NSB groups. In contrast, there was no significant difference in IL-1β (*p* > 0.05) among the three treatment groups. Furthermore, supplementation of the NSB did not show a significant difference (*p* > 0.05) in IL-6 cytokine and other blood immunoglobulins among the three treatment groups (Figure 1B).

### 3.3. The NSB Supplementation Altered Small Intestinal Histomorphology

Intestinal morphology plays a significant role in nutrient absorption by increasing the absorption efficiency of the intestine in chickens. In this study, the VH and ratio of VH/CD for the jejunum (*p* < 0.05) and ileum (*p* < 0.01) segments significantly increased in the NSB group compared to the CON and ANT groups (Figure 2A,B). However, CD showed no significant difference (*p* > 0.05) in the jejunum and ileum segments.

### 3.4. The NSB Supplementation Altered Intestinal Barrier and Tight Junction Proteins

The effects of NSB supplementation on intestinal barrier function and tight junction proteins are shown in Figure 3A,B. The expression of zonula occludens-1 (ZO-1) (*p* < 0.01), claudin-1 (*p* < 0.01), and occludin (*p* < 0.05) was significantly up-regulated in the NSB group in the jejunum (Figure 3A). Similarly, claudin-1 (*p* < 0.01) and mucin-2 (*p* < 0.05) were significantly up-regulated in the ileum supplemented with NSB compared to the CON and ANT groups. However, ZO-1 did not significantly increase (*p* > 0.05) in the ileum among the treatment groups (Figure 3B).

### 3.5. The NSB Altered Small Intestinal Antioxidant Capacity

Antioxidant capacity indicators such as SOD, T-AOC, and MDA were analyzed to determine if the NSB affects the intestinal oxidative stress of chickens using test kits. The results showed that SOD significantly increased in both the jejunum and ileum (*p* < 0.05), and the ratio of T-AOC/MDA in the jejunum (*p* < 0.01) for the NSB group compared to the CON and ANT groups (Figure 4A,B). Conversely, MDA significantly increased (*p* < 0.01) in the jejunum-fed ANT group compared to the CON and NSB groups (Figure 4A). Furthermore, the level of T-AOC in both the jejunum and ileum, as well as MDA and the ratio of T-AOC/MDA in the ileum, did not show a significant difference (*p* < 0.05) among the three treatment groups (Figure 4A,B).

### 3.6. The NSB Promoted Intestinal SCFA Production

The effect of NSB supplementation on the SCFAs’ content in the jejunum and ileum segments is also shown in Figure 5A,B. The results showed that NSB significantly increased (*p* < 0.01) the butyric acid content in the jejunum and ileum segments. Similarly, the level of propionic acid was significantly increased in the NSB group compared to the ANT but no significant changes with the CON group in both segments. However, there was no significant difference (*p* > 0.05) in the acetic acid content between the two intestinal segments among the groups.

### 3.7. 16S rRNA Sequencing, Composition, and Diversity of Cecal Microbiota

According to 16S rRNA gene sequencing, analysis of 24 cecal microbiota diversity samples was completed. A total of 1,700,004 optimized sequences were obtained for this study, with an amplified region of 338F_806R, a total length of 698,782,103 bases, and an average sequence length of 411.06 bp (base pair) per read. The estimated Good’s coverage for all cecal samples, which is used to assess sequencing depth, was 0.99, approaching 1.0, indicating that it was sufficient (Figure 6E). The species annotation results also showed the identification of 8 phyla, 14 classes, 37 orders, 60 families, 143 genera, 282 species, and 725 OUTs (operational taxonomic units) at a 97% similarity level.

The alpha diversity indexes, which measure the total richness and the effective number of bacterial species, showed no significant difference (*p* > 0.05) among the treatment groups at the OTU level (Figure 6A–D). The Venn diagram analysis also showed that 631 OTUs were commonly found in the three groups, while 17, 3, and 21 OTUs were also found in the CON, ANT, and NSB groups, respectively (Figure 6J). Higher OTU (25) levels were also found between the CON and NSB groups than the ANT group. The principal co-ordinate analysis (PCoA), which shows the associated cecal microbiota of broilers and reflects the beta diversity analysis, is presented in Figure 6K. Each group of cecal microbial communities clustered differently among the three groups.

### 3.8. Microbial Community Abundance at Phylum and Genus Levels

Relative community abundance of bacterial phyla showed that *Firmicutes*, *Bacteroidota*, *Actinobacteriota*, *Proteobacteria*, and *Cyanobacteria* were the top five phyla accounting for the highest percentage of cecal microbiota sequences in the three treatment groups (Figure 7A). Similarly, *Barnesiella*, *Faecalibacterium*, *norank_f__norank_o__Clostridia_UCG-014*, *Ruminococcus_torques_group*, *unclassified_f__Lachnospiraceae*, *Alistipes*, *norank_f__Eubacterium_coprostanoligenes_group*, *norank_f__norank_o__RF39*, *Bacteroides*, and *Lachnoclostridium* were the top ten bacterial genera (Figure 7B), accounting for the highest percentage of the cecal microbiota composition. No significant differences (*p* > 0.05) were found in the relative abundance of bacterial communities at the phylum and genus levels among the three treatment groups.

Linear discriminant analysis (LDA) (threshold > 2.0, *p* < 0.05) combined with LEfSe was used to identify significant differences in taxonomic composition among the treatment groups (Figure 7C,D). It is worth noting that *unclassified_f__Peptostreptococcaceae* (*p* < 0.05), *Colidextribacter* (*p* < 0.05), *norank_f__norank_o__norank_c__norank_p__Firmicutes* (*p* < 0.05), *Anaerostignum* (*p* < 0.05), GCA900066575 (*p* < 0.05), *Oscillospira* (*p* < 0.05), and *norank_f__Erysipelatoclostridiaceae* (*p* < 0.05) were identified in the NSB group, while *Shuttleworthia* (*p* < 0.05), *Anaeroplasma* (*p* < 0.05), and *Tyzzerella* (*p* < 0.05) were identified in the ANT group as a dominant taxon-enriched bacterial genera (Figure 7D). Furthermore, *Novosphingobium* (*p* < 0.05), *DTU089* (*p* < 0.05), *Enorma* (*p* < 0.05), and *Parabacteroides* (*p* < 0.05) were also identified as enriched bacterial genera in the CON group. It can be observed that the highest bacterial diversity is found in the NSB-supplemented group.

### 3.9. Correlation Analysis

We conducted a Spearman correlation analysis to see the possible relationships between cecal microbiota at the genus level with growth performance and other intestinal health indicators (Figure 8A). The results showed that *Shuttleworthia* was negatively correlated with SOD and IL-10 (*p* < 0.05) but positively correlated with MDA (*p* < 0.05) levels. *Colidextribacter* was positively correlated with ADFI, ADG, VH, and claudin-1 (*p* < 0.05). *Unclassified_f__Peptostreptococcaceae* was also positively correlated with ADFI, IL-10, and ZO-1 (*p* < 0.05). Lastly, *Enorma* was negatively correlated with ADG, claudin-1, and butyric acid but positively correlated with FCR (*p* < 0.05).

A Spearman correlation analysis was also conducted to examine the monotonic relationships between growth performance and other intestinal health indicators (Figure 8B). ADFI was positively correlated with IL-10, claudin-1, mucin-2, SOD, T-AOC, and butyric acid (*p* < 0.05). ADG was positively correlated with VH, IL-10, claudin-1, SOD, T-AOC, and butyric acid (*p* < 0.05). Furthermore, FCR was negatively correlated with VH, CD, and butyric acid (*p* < 0.05).

## 4. Discussion

In animal nutrition, SB is known to promote growth, regulate immunity, and maintain gut health. Our finding showed a significant impact of NSB on the performance of broilers, as evidenced by the significant changes in ADFI, ADG, and FCR during the rapid growth stage. This may be attributed to the effects of NSB on gut tissue and intestinal development, as well as its ability to enhance nutrient digestion and absorption, as indicated by the changes in FCR. Previous studies have reported variable results regarding the effects of different forms of protected SB on growth performance [8,15,16]. This study partially supports the findings of Wan et al. [9], who reported that chemically protected SB showed significant improvements. However, our finding disagrees with that of Wu et al. [11], who found no significant changes in all phases of broiler growth supplemented with protected SB. These observed differences may be attributed to variations in dosage, diet structure, release time of the product, microbiota composition, and the health status of the chickens.

Inflammatory cytokines play a crucial role in the immune response and inflammation, and serve as significant mediators of intestinal dysfunction. This study found that IL-10 was up-regulated and TNF-α was down-regulated in the NSB-supplemented group. IL-10 is an anti-inflammatory cytokine that regulates the immune system and helps maintain the integrity and balance of tissue epithelial layers [17]. TNF-α, on the other hand, is a pro-inflammatory cytokine produced and released by macrophages, and it serves as a key messenger that triggers metabolic changes following an innate immune response [18]. Yang et al. [19] found supportive results showing that 1000 mg/kg microencapsulated SB supplementation in the diet of broilers when challenged with *C. perfringens* decreased the level of TNF-α. Wan et al. [9] similarly demonstrated the positive effects of serum IL-10 in broilers fed chemically protected SB. Zou et al. [3] reported that SB can help maintain gut leakiness and decrease inflammatory gene expression. This study confirms that the NSB has a protective effect, which may be attributed to decreased pro-inflammatory stimulation and increased anti-inflammatory responses.

The histomorphological structure of chickens is the main indicator of healthy nutrient absorption and intestinal function [20]. A higher VH indicates proper functioning of the small intestine, while a higher VH/CD ratio reflects a stronger ability of the small intestine to digest and absorb nutrients. This suggests that a healthy intestinal morphology contributes to maintaining effective gut barrier function and permeability [21]. The current study found that NSB significantly increased the VH and VH/CD ratio in the jejunum and ileum of broilers. It is possible that NSB, when slowly released in the small intestine, helps in the development of intestinal villi and proper nutrient digestion. Previous studies have also mentioned the potential use of coated SB to improve the morphological structure during both the early and later growth stages [15,22]. Supportive findings were also reported in laying hens fed 800 mg/kg SB that showed a significant increment in the jejunum and ileum VH [23]. Furthermore, the increased levels of butyric and propionic acids in the jejunum and ileum contributed to the enhanced villus length and facilitated the proliferation of intestinal epithelial cells [24].

The intestinal barrier is essential for protecting the mucosal tissue by preventing the entry of harmful substances, thus being crucial for chickens, as it helps to maintain their intestinal environment [25]. ZO-1, occludin, and claudin-1 are the main intestinal tight junction proteins that form tight junctions between epithelial cells, protect the intestinal barrier, and restore gut barrier function [26]. Similarly, mucin-2 is also a major gel-forming mucin, representing a primary barrier component of mucus layers and supporting intestinal mucosal barrier integrity and immunological homeostasis [27]. In our study, we found that ZO-1, claudin-1, occludin, and to some extent mucin-2 in the jejunum, as well as claudin-1, occludin, and mucin-2 expression in the ileum, were up-regulated in the NSB-supplemented group. Supporting this study, Miao et al. [24] found that 500–750 mg/kg of protected SB supplementation affects the expression of occludin and ZO-1 in the jejunum, and claudin-1 and occludin in the ileum of laying hens. Similar findings in *necrotic enteritis*-challenged broilers supplemented with 800 mg/kg microencapsulated SB showed up-regulated occludin, claudin-1, mucin-2, and ZO-1 levels in the jejunum [28]. The dissociation of NSB to butyric acid in the intestine is crucial for stimulating intestinal epithelial cells and enhancing mucin synthesis. This, in turn, can lead to improved tight-junction integrity. There are multiple underlying mechanisms related to the up- or down-regulation of genes. SB is recognized for its ability to activate the interaction between the specificity protein 1 (SP1) transcription factor and the claudin-1 promoter, ultimately increasing protein abundance [29]. SB also activates adenosine monophosphate (AMP)-activated protein kinase (AMPK) through calcium/calmodulin-dependent protein kinase β (PKB) in the Caco-2 cells [30,31] and regulates the ATP substrate levels to accelerate the abundance of tight junction proteins in the intestines of animals [32]. Numerous studies, both in disease-challenged and unchallenged conditions [7,19] have demonstrated a positive response to protected SB in regulating intestinal barrier function. These findings support the observed positive response in the growth performance of chickens, as reported in this study.

Modern high-yielding chicken strains are susceptible to excessive reactive oxygen species (ROS) produced by the metabolism of various body cells, leading to intestinal oxidative stress [33]. SOD, T-AOC, and MDA are the key indicators for intestinal oxidative stress in chickens examined in this study. SOD plays an important role in the biological oxidant balance and protective function for intestinal development by eliminating free radicals produced by MDA. Our study shows that the content of SOD and T-AOC/MDA ratio in the jejunum, as well as SOD in the ileum, were significantly increased in the NSB group. In support of this study, Miao et al. [34] showed that coated SB at 500 g/kg increased SOD and decreased MDA activities in laying hens. De-Cara et al. [35] also found that supplementation of protected SB at 2 g/kg increased SOD activity in broilers. Similarly, Zhang et al. [36] found that microencapsulated SB at 400 mg/kg decreased the level of MDA content, indicating that NSB has a protective role in alleviating oxidative stress in birds. However, the effects of salt-protected SB on intestinal antioxidant capacity, gut barrier function, and tight junction proteins in broilers are limited in the literature, except for studies by Wu et al. [11] of broilers at 21 and 42 days, and by Zhao et al. [8] and by Mátis et al. [7] of broilers at 21 days, and need further in-depth investigation.

SCFAs’ content is considered the best indicator of a healthy gut flora population, which plays important roles in energy metabolism, immune function, and intestinal growth [37]. Supplementation of NSB provides slow dissociation in the intestinal segments and is expected to increase hindgut SCFA production. In this study, the significant increment in propionic and butyric acid in the jejunum and ileum in the NSB group indicates the improved changes in intestinal morphology and growth performance. Similar results were also observed in laying hens fed 800 mg/kg SB with increased butyric acid content in the ileum segment [23]. The increased SCFAs’ concentration in gut segments of poultry promotes immunological response, controls pathogenic bacteria, and affects intestinal morphology and function [38].

Cecum microbiota can influence immune function by interacting with the intestinal epithelium through gut metabolites and maintaining gut health [39]. Although there were no significant changes in alpha diversity or relative community abundance among the groups, the NSB group had a higher percentage of *Firmicutes* abundance at the phylum level. This finding aligns with Miao et al. [34], who found a similar result in laying hens fed coated SB. *Firmicutes* is the most bacterial phylum found in the healthy intestines of chickens. It decomposes dietary fibers, leading to increased production of SCFAs, including butyrate. The intestinal epithelial cells then preferentially absorb butyrate as an energy source, subsequently promoting the proliferation of epithelial cells and maintaining gut integrity and barrier function [40] through the molecular mechanisms mentioned earlier. Likewise, LEFSe analysis showed a significant change in *Peptostreptococcaceae*, *Firmicutes*, *Oscillospira*, *Erysipelatoclostridiaceae*, and others (Figure 7D) in the NSB group. *Peptostreptococcaceae* is typically considered a normal commensal bacteria, and its proportion is higher in the gut microbiota of healthy animals than in those with dysbiosis, indicating that this bacteria contributes to gut homeostasis [41]. Supportive to this finding, Vieira et al. [42] on broilers and Hao et al. [43] on white pekin laying ducks showed that *Peptostreptococcaceae* and *Erysipelatoclostridiaceae* are butyric acid-producing bacteria used for efficient nutrient absorption and stimulating intestinal mucosal cell growth, demonstrating the improved changes in the VH and VH/CD ratio in the NSB group. Additionally, *Oscillospira* is probably a genus capable of producing SCFAs such as butyrate and has a positive effect on gut microbiota [44]. More importantly, the change in cecal microbiota abundance at the phylum and genus levels resulted from regulating microbes and nutrients by NSB supplementation, which finally up-regulated the intestinal barrier gene expression by altering cecum microbial metabolism [2]. Furthermore, SCFAs produced by these bacterial species promote mucin-2 expression, which is essential to maintaining intestinal mucosa and regulating the microbial community structure of chickens [38]. In general, all the above gut metabolites may contribute to the changes in butyric and propionic acid levels observed in the NSB group in this study.

The Spearman correlation analysis of gut metabolites with growth performance and other intestinal health indicators showed a positive correlation, mainly with *Colidextribacter* and *Peptostreptococcaceae* as shown in Figure 7A. The genus *Colidextribacter*, belonging to the phylum *Firmicutes*, promotes SCFA production, thus reducing intestinal inflammation and maintaining gut mucosal integrity [45]. Similarly, *Peptostreptococcaceae* is also under the family of *Clostridiaceae*, most abundant in the cecum of birds, and known for SCFA production from dietary fibers [46]. In this study, the positive correlation between *Colidextribacter* with ADG, VH, and claudin-1 also indicates that this metabolite promotes the immune response and maintains mucosal gut integrity, contributing to growth and gut health in broilers [47]. The positive correlation of ADFI and ADG and the negative correlation of FCR with other intestinal health indicators in this study (Figure 7B) also indicate the positive response of NSB to promoting the growth performance and health of broilers, showing that NSB has the potential to replace antibiotics in poultry nutrition.

## 5. Conclusions

In this study, NSB supplementation improved ADFI, ADG, VH, the VH/CD ratio, IL-10 cytokine, ZO-1, claudin-1, SOD, the T-AOC/MDA ratio, propionic acid, butyric acid, and SCFA-producing bacteria such as *Peptostreptococcaceae*, *Colidextribacter*, *Oscillospira*, and *Erysipelatoclostridiaceae*. In conclusion, NSB significantly improved the intestinal histomorphology, gut barrier function, oxidative capacity, and microbiota community, contributing to the enhanced growth performance and overall intestinal health of broilers. This shows NSB can be used as a potential feed additive in the broiler industry. Further studies may be required to examine the effects of NSB on disease-challenged chicken models and in different environmental conditions.

## Figures and Tables

**Figure 1 biology-13-00317-f001:**
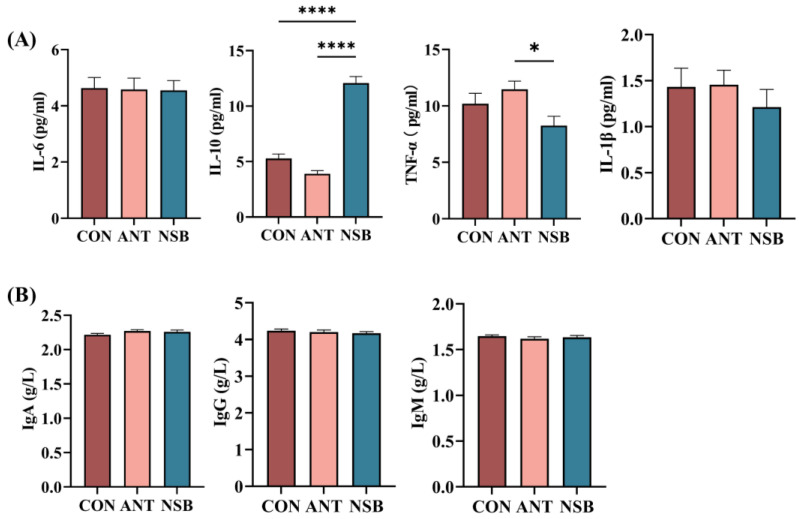
Effects of NSB on serum cytokine expression and blood immunoglobulins of broilers. (**A**). Serum pro-inflammatory cytokines. (**B**). Blood immunoglobulins. CON = control (corn–soybean meal basal diet); ANT = antibiotic diet (basal diet + 100 mg/kg aureomycin); NSB = new buffer salt-protected sodium butyrate (basal diet + 800 mg/kg NSB). * *p* < 0.05, **** *p* < 0.0001, *n* = 8.

**Figure 2 biology-13-00317-f002:**
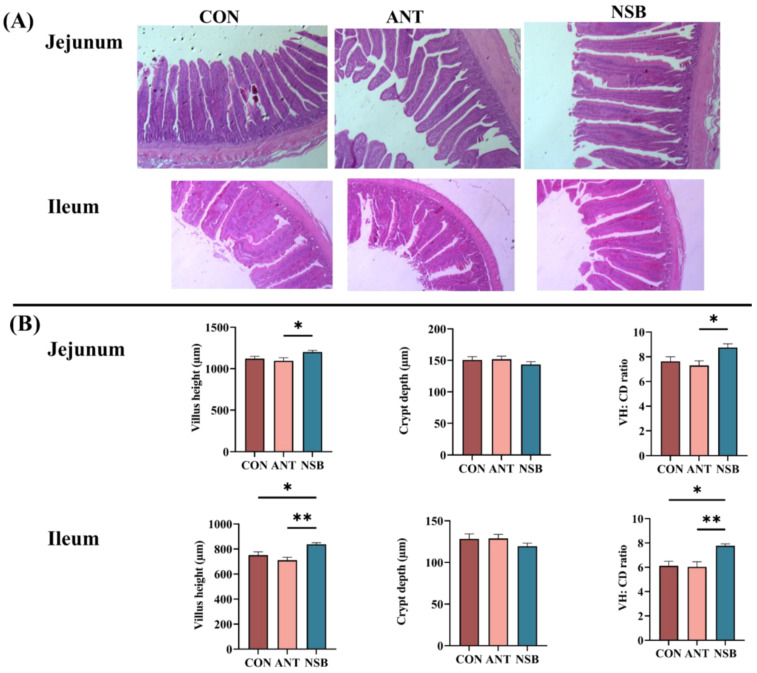
Effects of NSB on jejunum and ileum morphology of broilers. (**A**). H&E representative staining image in the jejunum and ileum. (**B**). VH, CD, and VH/CD ratio of the jejunum and ileum. CON = control (corn–soybean meal basal diet); ANT = antibiotic diet (basal diet + 100 mg/kg aureomycin); NSB = new buffer salt-protected sodium butyrate (basal diet + 800 mg/kg NSB). * *p* < 0.05, ** *p* < 0.01, *n* = 8.

**Figure 3 biology-13-00317-f003:**
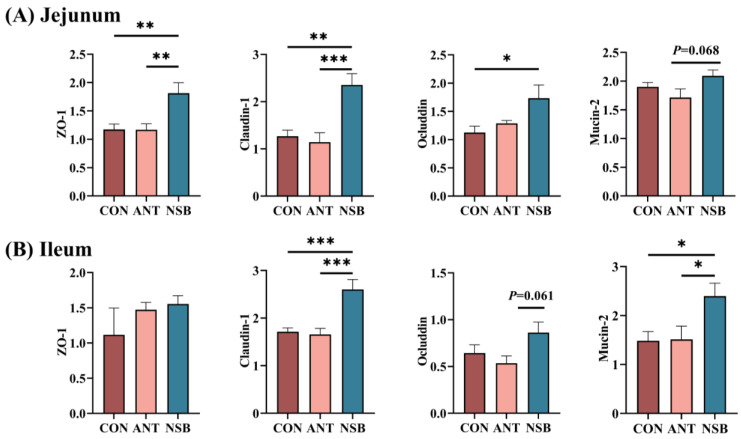
Effects of NSB on mRNA expressions of jejunum (**A**) and ileum (**B**) mucosal tissues. CON = control (corn–soybean meal basal diet); ANT = antibiotic diet (basal diet + 100 mg/kg aureomycin); NSB = new buffer salt-protected sodium butyrate (basal diet + 800 mg/kg NSB). * *p* < 0.05, ** *p* < 0.01, *** *p* < 0.001, *n* = 8.

**Figure 4 biology-13-00317-f004:**
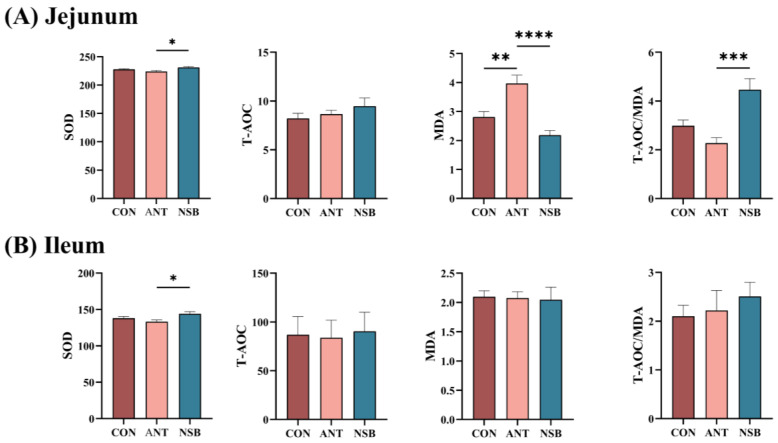
Effects of NSB on the jejunum (**A**) and ileum (**B**) antioxidant capacity of broilers. CON = control (corn–soybean meal basal diet); ANT = antibiotic diet (basal diet + 100 mg/kg aureomycin); NSB = new buffer salt-protected sodium butyrate (basal diet + 800 mg/kg NSB). * *p* < 0.05, ** *p* < 0.01, *** *p* < 0.001, **** *p* < 0.0001, *n* = 8.

**Figure 5 biology-13-00317-f005:**
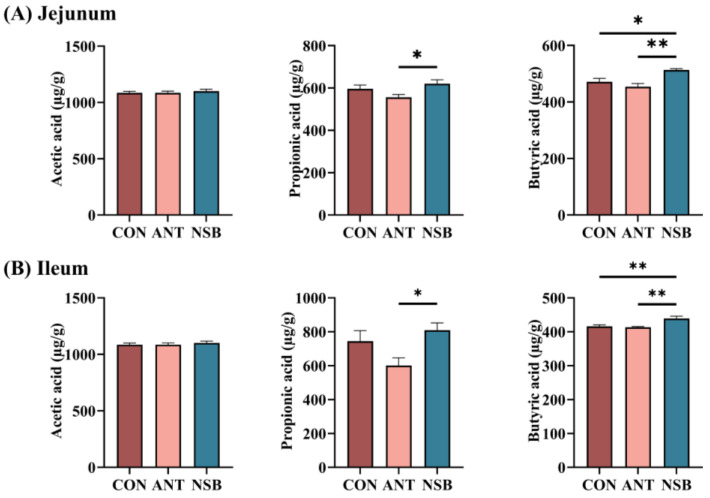
Effects of NSB on SCFAs’ content in the jejunum (**A**) and ileum (**B**) segments. CON = control (corn–soybean meal basal diet); ANT = antibiotic diet (basal diet + 100 mg/kg aureomycin); NSB = new buffer salt-protected sodium butyrate (basal diet + 800 mg/kg NSB). * *p* < 0.05, ** *p* < 0.01, *n* = 8.

**Figure 6 biology-13-00317-f006:**
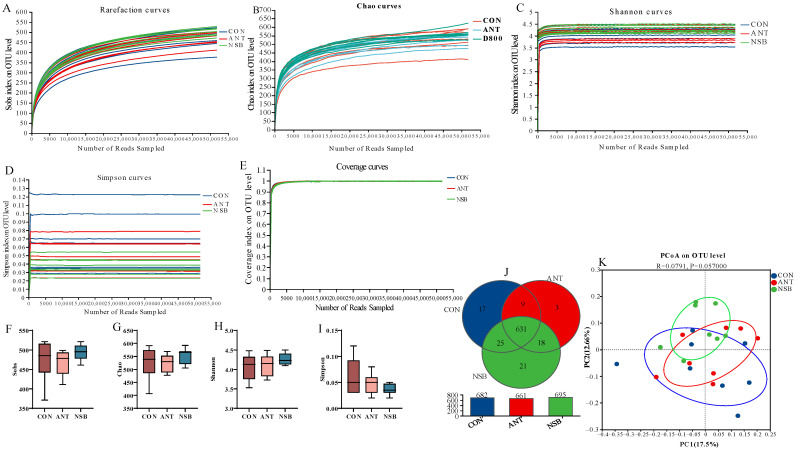
Shows the refraction curves that reflect the rationality of the samples (**A**–**E**). Alpha diversity indexes (**F**–**I**) show the richness and evenness of cecal microbiota diversity. Venn diagram (**J**) shows the number of bacterial species or taxa that are shared among the three groups and the principal co-ordinate analysis (PCoA) based on the Bray–Curtis ANISOM test (**K**). CON = control (corn–soybean meal basal diet); ANT = antibiotic diet (basal diet + 100 mg/kg aureomycin); NSB = the new buffer salt-protected sodium butyrate (basal diet + 800 mg/kg NSB), *n* = 8.

**Figure 7 biology-13-00317-f007:**
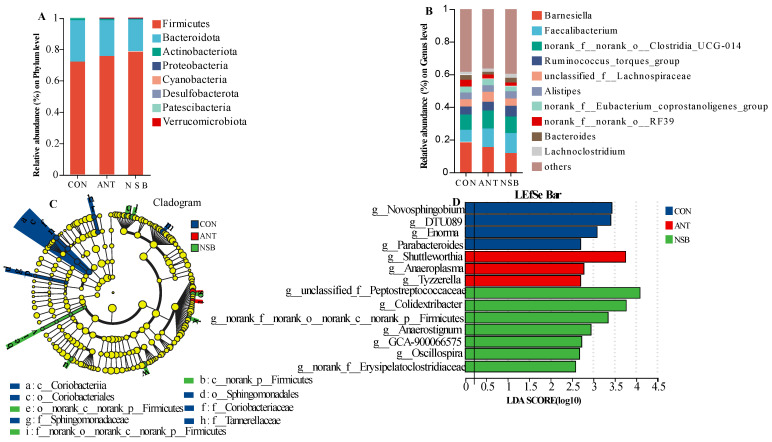
Relative community abundance of cecal microbiota at phylum (**A**) and genus (**B**) levels. Linear discriminant analysis effect size (LEfSe) cladogram shows different cecum microbial taxa from phylum to family level (**C**). The histogram shows taxonomical classification of cecal microbes at the genus level (LDA value, ≥2.0, *p* ≤ 0.05) (**D**). CON = control (corn–soybean meal basal diet); ANT = antibiotic diet (basal diet + 100 mg/kg aureomycin); NSB = soyabean (basal diet + 800 mg/kg NSB), *n* = 8.

**Figure 8 biology-13-00317-f008:**
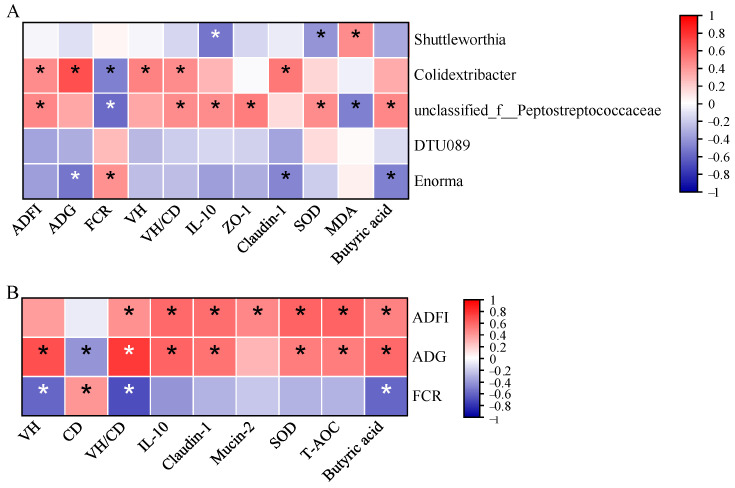
Spearman correlation analysis between cecal microbiota at genus level with growth performance and intestinal health indicators in the jejunum (**A**). Spearman correlation analysis between growth performance with other intestinal health indicators (**B**). The red and blue colors indicate positive and negative correlations, respectively. * Means *p* < 0.05, *n* = 8.

**Table 1 biology-13-00317-t001:** Composition and nutrient contents of basal diet in broilers.

Item	Experimental Basal Diet
Corn	61.00
Soybean meal	31.80
Soybean oil	2.90
Wheat middling	0.97
Limestone	1.01
Salt	0.35
Calcium dihydrogen phosphate	1.39
Choline chloride	0.20
Methionine	0.12
Lysine	0.01
Vitamin premix ^1^	0.05
Mineral premix ^2^	0.20
Total	100.00
Nutritional values ^3^	
NE (Mcal/kg)	3.05
CP (%)	19.50
Calcium (%)	0.80
Available phosphorus (%)	0.38
Lysine (%)	1.05
Methionine (%)	0.42

^1^ Supplies the following per kg diet: Vitamin A, 9140 IU; Vitamin D3, 4405 IU; Vitamin E, 11 IU; water-soluble Vitamin K, 7.30 mg; riboflavin, 9.15 mg; D-pantothenic acid, 18.33 mg; *Niacin*, 73.50 mg; choline chloride, 1285 mg; Vitamin B12, 200 µg; biotin, 900 ug; *Thiamine nitrate*, 3.67 mg; folic acid, 1650 ug; pyridoxine hydrochloride, 5.50 mg. ^2^ Supplies the following per kg diet: I, 1.85 mg; Mn, 110.10 mg; Cu, 7.40 mg; Fe, 73.50 mg; Zn, 73.50 mg; Se, 500 ug. ^3^ NE, CP, calcium, and available phosphorus were determined by routine methods. Lysine and methionine were detected by an automatic amino acid analyzer. Each diet was measured for six replicates.

**Table 2 biology-13-00317-t002:** Effects of supplementation of the new buffer salt-protected sodium butyrate on the growth performance of broilers (*n* = 8).

Parameters	Dietary Treatments			SEM	*p*-Value
	CON	ANT	NSB		
ADG (g/day/bird)	87.02 ^b^	87.52 ^b^	91.00 ^a^	0.44	*p* < 0.001
ADFI (g/day/bird)	157.03 ^b^	158.02 ^b^	161.17 ^a^	0.46	*p* < 0.001
FCR (feed/gain ratio)	1.80 ^a^	1.80 ^a^	1.76 ^b^	0.005	0.003

ADFI = average daily feed intake; ADG = average daily gain; FCR = feed conversion ration; CON = control (corn–soybean meal basal diet); ANT = antibiotic (basal diet + 100 mg/kg aureomycin); NSB = new buffer salt-protected sodium butyrate (basal diet + 800 mg/kg NSB); SEM = standard error of mean. Superscripts (a, b) in the same row indicate significant differences (*p* < 0.01).

## Data Availability

Data are contained within the article. The raw sequencing reads data were deposited in the NCBI Sequence Read Archive (SRA) database (Accession Number: PRJNA1025322).

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
