# Peer review of "The New Buffer Salt-Protected Sodium Butyrate Promotes Growth Performance by Improving Intestinal Histomorphology, Barrier Function, Antioxidative Capacity, and Microbiota Community of Broilers"

_biology, 2024, doi:10.3390/biology13050317_

Round 1
Reviewer 1 Report
Comments and Suggestions for Authors
Dear Editor and Authors,
In this study, the authors aimed to clarify the effectiveness of a new buffer salt-protected sodium butyrate as a growth promoter compared to antibiotics. Employing a comprehensive methodology, they examined the impact of this compound on the gut histomorphology, barrier function, antioxidative capacity, and microbiota community of broilers. In the introduction, the authors provide a clear overview of the subject and articulate their objectives explicitly. The material method section is clear and understandable. They presented their own conclusions in a clear and comprehensible manner, supporting their arguments with visual support.
The Discussion section is comprehensive and detailed. However, it lacks in-depth exploration of the possible mechanisms underlying the upregulation or downregulation of genes, which is typically expected in studies involving expression. Furthermore, the impact of microbiota, particularly at the phylum and genus levels, on these mechanisms is not adequately addressed. Addressing these deficiencies would enhance the interest and impact of the current article.
Best regards
Minor comments:
L22: Kindly write the clear version, as it is only referenced once in the simple summary.
L26-28, L174-175: It is usually advised not to begin a sentence with a numerical value. Kindly attempt to rephrase your statement.
L30-51: Kindly refer to the clear definition of the term prior to using abbreviations in the abstract. This portion holds significance in facilitating a wider audience reach.
L60: This situation involves numerous countries; therefore, it is preferable not to restrict it to a single country.
L69: Kindly write the clear version siince it is used for the first time.
L72: Please write “Mátis et al. [7]” instead of “Mátis et al.”.
L94, L100, L393: Please write “NSB” instead of “buffer salt-protected sodium butyrate (NSB)” since it has been already abbreviated.
L120, L123: Please write “diet” instead of “DIET”.
L130: The duration (in minutes) and “g” or “rpm” that the centrifuge was set for in order to extract the serum should to be specified.
L145: Which manufacturer's instructions? Kindly provide the manufacturer's name to whom the catalog number was provided.
L157: Please write “RT-qPCR” instead of “RT-QPCR”.
L172: Please provide sources.
L189: Please use "including" instead of "such as". It will be understood that you have performed additional SCFAs analyses if you use the phrase "such as."
L190: The expression "mother liquor" is incorrect. Kindly employ appropriate terminology.
L239-240: This sentence necessitates alteration. The intended characteristic of an additive is to reduce the FCR, or alternatively, to improve it. However, the existing phrasing indicates that CON and ANT are more effective. A different phrase would be useful.
L260, L272, L312: Specifying the “Means” is unnecessary and can be advantageous to remove.
L265: Please write “CD” instead of “crypt depth”.
L314: Please check the spelling of the subheading.
L72, L80, L109, L117, L391, L401, L403, L418, L431, L433, L454, L464, L466, L488: Please write “SB” instead of “sodium butyrate”.
L415: Please write “Yang et al. [19]” instead of “Yang et al.”.
L424: Please write “VH” instead of “villus height”.
L495: Please write “Vieira et al. [37] and Hao et al. [38]” instead of “Vieira et al. And Hao et al.”.
Author Response
Please see the attached Word document. Thank you very much.

Reviewer 2 Report
Comments and Suggestions for Authors
Comments to the Authors
The current manuscript provides detailed information on the growth performance, serum anti-inflammatory cytokines, intestinal morphology, gut barrier function, antioxidative parameters, SCFAs content, and cecal microbiota in broiler fed diet supplemented with a commercial sodium butyrate protected by a new buffer salt solution (NSB) used as an antibiotic alternative. The aim of this study was to investigate how the new buffer salt-protected sodium butyrate affects growth performance, serum cytokines and immunoglobulins, histomorphology, intestinal immunity and oxidative capacity, SCFAs content, and cecal microbiota during the rapid developmental stage of broilers. The authors of this study showed that the NSB improves growth performance, serum anti-inflammatory cytokines, gut morphology, intestinal immunity and antioxidant capacity, SCFAs content, and cecum microbiota in broilers.The authors have done a commendable job of substantiating their claims with appropriate methodology. The manuscript is well illustrated with clear, well-detailed methods. This study is linked to the United Nations’ sustainable development objectives, helping to tackle some of the world's greatest challenges. The manuscript could be considered for publication after addressing the following shortcomings.
Line 99: to be logical, I suggest “Approval” instead of “approval”.
Line 105: I suggest “A total of 192 1-day-old Arbor Acres (AA)…” instead of “A total of one 192 1-day-old Arbor Acres (AA)…”.
Line 117: I suggest “kg” instead of “Kg”.
Line 128: I suggest “At 42-days-old,…” instead of “At 42 days of age”
Line 228 and 365: I suggest “p < 0.05” instead of “P < 0.05”.
Line 314: I suggest “3.7. 16S rRNA” instead of “3.7.16. S rRNA”.
Line 369-370: The sentence “Cecum microbiota can influence immune function by interacting with the intestinal epithelium through gut metabolites and maintaining gut health [14]” must be in the discussion section.
Line 392-395: Sentence “This study evaluated the effects of a new slow-release buffer salt-protected sodium butyrate (NSB) on the growth performance, serum cytokines and immunoglobulins, intestinal morphology, intestinal barrier function, antioxidant capacity, SCFAs, and cecal microbiota of broilers.” should be deleted because the objective has already been given in the introduction.

Author Response

(The authors gave the same response as above.)

Round 2
Reviewer 1 Report
Comments and Suggestions for Authors
Dear Editor and Authors,
Please see the attached file.

Author Response
Please see the attachment below. Thank you very much.
